# Post-Irradiation Hyperamylasemia Is a Prognostic Marker for Allogeneic Hematopoietic Stem Cell Transplantation Outcomes in Pediatric Population: A Retrospective Single-Centre Cohort Analysis

**DOI:** 10.3390/jcm10173834

**Published:** 2021-08-26

**Authors:** Francesco Baldo, Roberto Simeone, Annalisa Marcuzzi, Antonio Giacomo Grasso, Rossella Vidimari, Francesca Ciriello, Davide Zanon, Alessandra Maestro, Egidio Barbi, Natalia Maximova

**Affiliations:** 1Department of Medicine, Surgery and Health Sciences, University of Trieste, Piazzale Europa 1, 34127 Trieste, Italy; francesco.baldo@burlo.trieste.it (F.B.); egidio.barbi@burlo.trieste.it (E.B.); 2Department of Transfusion Medicine, ASUGI, Piazza dell’Ospitale 1, 34125 Trieste, Italy; roberto.simeone@asugi.sanita.fvg.it; 3Department of Chemical and Pharmaceutical Sciences, University of Ferrara, via Ludovico Ariosto 35, 44121 Ferrara, Italy; annalisa.marcuzzi@unife.it; 4Institute for Maternal and Child Health-IRCCS Burlo Garofolo, via dell’Istria 65/1, 34137 Trieste, Italy; antonio.grasso@burlo.trieste.it (A.G.G.); davide.zanon@burlo.trieste.it (D.Z.); alessandra.maestro@burlo.trieste.it (A.M.); 5Department of Medical Physics, ASUGI, via Pietà 2/2, 34129 Trieste, Italy; rossella.vidimari@asugi.sanita.fvg.it; 6Department of Radiotherapy, ASUGI, via Pietà 19, 34129 Trieste, Italy; francesca.ciriello@asugi.sanita.fvg.it

**Keywords:** total body irradiation, total amylase, proinflammatory cytokines, hematopoietic stem cell transplantation, overall survival, leukemia relapse, pediatric patients

## Abstract

Background: Total body irradiation (TBI) is a mandatory step for patients with acute lymphoblastic leukemia (ALL), undergoing allogeneic hematopoietic stem cell transplantation (HSCT). In the past, amylases have been reported to be a possible sign of TBI toxicity. We investigated the relationship between total amylases (TA) and transplant-related outcomes in pediatric recipients. Methods: We retrospectively analyzed the medical records of all the patients who underwent allogeneic HSCT between January 2000 and November 2019. The inclusion criteria were the following: recipient’s age between 2 and 18, diagnosis of ALL, no previous transplantation, and use of TBI-based conditioning. The serum total amylase and pancreatic amylase were evaluated before, during, and after transplantation. Cytokines and chemokines assays were retrospectively performed. Results: 78 patients fulfilled the inclusion criteria. Fifty-seven patients were treated with fractionated TBI, and 21 with a single-dose regimen. The overall survival (OS) was 62.8%. Elevated values of TA were detected in 71 patients (91%). The TA were excellent in predicting the OS (AUC = 0.773; 95% CI = 0.66–0.86; *p* < 0.001). TA values below 374 U/L were correlated with a higher OS. The highest mean TA values (673 U/L) were associated with a high disease-progression mortality rate. The TA showed a high predictive performance for disease progression-related death (AUC = 0.865; 95% CI = 0.77–0.93; *p* < 0.0001). Elevated TA values were also connected with significantly higher levels of proinflammatory cytokines, such as TNF-α, IL-6, and RANTES (*p* < 0.001). Conclusions: this study shows that TA is a valuable predictor of post-transplant OS and increased risk of leukemia relapse.

## 1. Introduction

Allogeneic hematopoietic stem cell transplantation (HSCT) is a highly specialized medical procedure that, nearly 60 years ago, introduced the first regenerative approach to clinical practice [1,2]. Although HSCT technology has evolved considerably in the recent years, total body irradiation (TBI) remained one of the leading conditioning regimens in pediatric and adult patients with acute lymphoblastic leukemia (ALL) [3,4].

In the late 1970s, the TBI procedure underwent a radical change, passing from single high-dose administrations to fractionated regimens [5], remaining substantially unchanged to the present day, although methodological differences still exist between centers and countries [6].

TBI has significant advantages over high-dose chemotherapy, including accessibility to “sanctuaries” sites, such as the testicles and central nervous system, homogeneity of high doses throughout the body, absence of concerns for drug excretion or detoxification, and cross-resistance in combination with chemotherapy, as well as the capacity to shield or boost the sites of interest [6].

However, TBI is responsible for many significant side effects, such as veno-occlusive disease (VOD), renal toxicity, interstitial pneumonitis, secondary malignancies, reproductive insufficiency, as well as growth retardation 3. Their incidence has dramatically reduced after introducing fractionated regimens and decreasing dose rates [7,8,9].

In recent years, a great deal of effort has been made to reduce TBI conditioning’s toxicity, and identify people who are at higher risk of developing complications, aiming for personalized radiotherapy [10,11,12], including the identification of molecular biomarkers that might predict the response and tolerability of this procedure [10,11,12,13]. Increased amylase levels, both pancreatic and salivary, and clinical manifestations of acute parotitis or pancreatitis, have been reported during TBI-based conditioning [14,15,16]. We decided to investigate the possible relationship between TA values and our cohort of patients’ transplant outcomes.

## 2. Materials and Methods

### 2.1. Study Population

A retrospective single-center cohort analysis was conducted at the Pediatric Bone Marrow Transplant Unit of the Institute for Maternal and Child Health “IRCCS Burlo Garofolo” in Trieste, Italy. The Institutional Review Board of the institute (reference IRB-BURLO no. 03/2020) approved the study protocol. The parents of all subjects enrolled signed written consent for the collection and use of their personal data. Medical records of 226 patients who underwent allogeneic HSCT between January 2000 and November 2019 were investigated.

### 2.2. Inclusion and Exclusion Criteria

Inclusion criteria were the following: subjects between 2 and 18 years of age at the time of HSCT, diagnosis of ALL, no history of previous transplants, myeloablative conditioning regime including high-dose TBI, documented serum TA, and pancreatic α-amylase levels before and during TBI-based conditioning. Patients were excluded if they had documented abnormal TA and pancreatic α-amylase levels one month before HSCT.

### 2.3. Study Predictors

We collected data about the demographic characteristics of the patients, type of donor, and status of the disease at transplantation. Disease risk was defined as low for patients in first complete remission (CR1) and high risk in all further complete remissions or non-complete remissions (non-CR). We gathered data on overall survival (OS), with which we describe the number of patients that survived after TBI-based conditioning HSCT until the latest follow-up. We also analyzed the cumulative incidence of death, early (100 days) post-transplant complications, such as TBI-related organ toxicity, infections, neutrophil and platelets engraftment, and acute graft-versus-host disease (GVHD). We also collected data on late complications, including chronic organ injuries, such as chronic GVHD, hypothyroidism, pancreatic insufficiency, cataract, and growth hormone deficiency. Adverse events were categorized through the common terminology criteria for adverse events (CTCAE) by the National Cancer Institute [17]. The causes of death resulted in leukemia relapse and transplant-related mortality (TRM). We included GVHD, systemic infection, and organ toxicity in the type of outcomes defined as TRM.

### 2.4. HSCT Procedure and TBI Treatment

All 78 pediatric patients affected by ALL received high-dose TBI-based standard myeloablative conditioning for an allogeneic HSCT. In patients under the age of 2, TBI was omitted, according to the national protocol for ALL of the Italian Association of Pediatric Hematology and Oncology (AIEOP). We defined the following two TBI protocol groups: the first received a standard dose of 12 Gy, delivered in 6 fractions; the second one received 7.5 Gy in a single dose. A linear, accelerator-based, latero-lateral irradiation was employed for both the procedures. Plexiglass slabs were used to compensate for missing tissue in the head and neck, lead tablet in the lower leg regions, and lung shielding with the lateral position’s upper limbs. In vivo dosimetry was performed with thermoluminescence dosimetry (TLD) only until 2003. Subsequently, the double-check of delivered dose and dose homogeneity was performed with Gafchromic EBT3 film and MOSFET (metal oxide semiconductor field effect transistor) detectors.

The conditioning regimen and GVHD prophylaxis were conducted as previously described [17].

### 2.5. Total Amylase and Pancreatic α-Amylase Values Assessment

Serum TA was evaluated before, during, and after the TBI treatment. Baseline TA values were obtained immediately before the beginning of the conditioning regimen. TA values were obtained during the conditioning either after the first day of TBI, if patients underwent the single-dose protocol, or before the third TBI session, if they underwent the fractioned protocol. TA levels were measured daily until HSCT infusion or until normalization. We assessed the mean value and the highest measured value from baseline to HSCT (Figure 1). With the total amylase assessment, we analyzed the pancreatic α-amylase values to prove that the post-TBI total amylase raise was caused by an increase in the α-amylase only. Amylase levels were expressed in unity per liter (U/L). Total amylase levels above 100 U/L were considered abnormal.

### 2.6. Assessing TBI-Related Inflammatory Status

The analysis of 27 cytokines and chemokines, namely, IL-1β, IL-1ra, IL-2, IL-4, IL-5, IL-6, IL-7, IL-8, IL-9, IL-10, IL-12(p70), IL-13, IL-15, IL-17, eotaxin, FGF basic, G-CSF, GM-CSF, IFN-γ, IP-10, MCP-1(MCAF), MIP-1α, PDGF-bb, MIP-1β, RANTES, TNF-α, and VEGF, was carried out on plasma samples with multiple immunoassays, using a bead-based magnetic sensor (27 human Bio-Plex assay) (BIO-RAD Laboratories, Milan, Italy) following the instructions provided by the manufacturer. Data concerning reactions were acquired by a Bio-Plex 200 reader, while a digital processor and Bio-Plex Manager^®^ 6.0 software converted data into median fluorescence intensity and concentration (pg/µL) (BIO-RAD Laboratories, Milan, Italy) [18]. We only considered the cytokine values determined at the end of TBI treatment. TA values were eventually related to the inflammatory cytokine levels.

### 2.7. Follow-Up

After discharge, patients were followed up monthly in the first six months and then every 6 to 12 months, in case of an uneventful post-transplant path. Follow-up duration was calculated from the date of HSCT to that of the patient’s last visit or death. A minimum of 1-year follow-up for survivors was considered.

### 2.8. Endpoints

The primary endpoint was to find a correlation between the maximum TA value during TBI treatment and OS, as well as other transplant outcomes. The outcomes were reported at the early and the late phases after HSCT. The post-transplant time phases were previously defined [19]. Toxicity was graded according to National Cancer Institute common toxicity criteria [20]. The secondary endpoint was to determine a TA cut-off value that could better predict adverse transplant outcomes.

### 2.9. Statistics

Descriptive statistics were used to determine the distribution and frequency of the variables. Continuous variables were expressed as median and confidence interval (CI) between second and third quartiles (percentile 25 and percentile 75) or mean ± standard deviation when appropriate, while categorical variables were expressed as frequency and absolute or a percentage value. Box-and-whisker plots were generated to display the numeric variables’ distribution. Mann–Whitney U test was performed to compare numeric variables between two different groups of patients. Kruskal–Wallis test was used for multiple comparisons between more than two groups. Fisher’s test was adopted to analyze categorical variables in different groups of patients. We investigated the TA’s validity in predicting transplant outcomes by assessing their respective area under the curve (AUC) and receiver operating characteristic (ROC) curve. Youden index was employed to establish the best cut-off for the sensitivity and specificity of each variable. Kaplan–Meier plots were generated for a graphical explanation of clinical outcomes, and log-rank test was used to compare survival curves. The *p*-values < 0.05 were considered statistically significant. Statistical analyses were performed using WinStat (v. 2012.1; In der Breite 30, 79189 Bad Krozingen, Germany) and MedCalc (Statistical Software version 18.9.1, Ostend, Belgium; http://www.medcalc.org; accessed on March–April 2021).

## 3. Results

### 3.1. Patients

Two hundred and nine patients underwent an allogeneic HSCT from January 2008 to November 2019. Seventy-eight of them had ALL and underwent TBI-based conditioning before HSCT, thus becoming eligible for the study. The median follow-up was 4.3 years (range 1.08–12.83 years). The patients’ and transplants’ features are shown in Table 1.

The OS after HSCT at the last follow-up was 62.8% (49 patients). No demographic or transplant-related predictors, such as gender, age at transplant, donor type, graft source, TBI protocol, or TBI-associated chemotherapy, were related to the OS.

Fifty-seven patients (73.1%) received TBI with fractioned doses of 12 Gy (standard protocol), while 21 patients (26.9%) took a single TBI dose of 7.5 Gy. The mean dose rate ± standard deviation (SD) was 14.0 ± 2.0 cGy/min in the 7.5 Gy group and 18.7 ± 1.7 cGy/min in the 12 Gy group. The mean percentage variation ± SD in dosimetry was 1.5 ± 1.0% and −0.9 ± 1.9% in the two groups, respectively, which was under the acceptable 10% range [5].

Abnormal levels of TA during the TBI treatment were found in 71 patients (91%), and the highest values of TA were observed after the first day of TBI. The maximum TA value was 2210 U/L, which was documented before the third session. The mean TA value after the first day of TBI was 341 U/L ± 344 U/L, while the median value was 246 U/L.

Comparing the maximum TA values in the patients who underwent the two different TBI protocols, assessed before the third radiotherapy session, did not reveal statistically significant differences (*p* = 0.2111). Indeed, the mean TA value in the 7.5 cGy group was 305 U/L (range 46–2210 U/L), while in the 12 cGy group, it was 223 U/L (range 25–1337 U/L).

### 3.2. Relationship between Maximum TA Values and OS

The maximum TA values predicted the post-transplant OS (AUC = 0.773; 95% CI = 0.66–0.86; *p* < 0.001). The maximum value of the Youden index was 374 U/L, with a corresponding sensitivity of 58.6% and specificity of 96%. The relationship between OS and maximum TA values is displayed in the corresponding ROC curve (Figure 2A).

Establishing an arbitrary cut-off of 374 U/L, the OS was 78% versus 11% for patients below this value versus 11% for those above it (*p* < 0.0001) (Figure 2B).

### 3.3. Relationship between Maximum TA Values, Status of Disease at Transplant and Causes of Death

We analyzed the causes of death in the study population. The all-cause mortality was 37.2% (29 patients). Comparing the stage of the disease at the transplant, as expected, most deaths were in the high-risk group (23 patients, 70.3%) compared to the low-risk group (6 patients, 20.7%). The mean TA was significantly higher in the deceased patients in both high- and low-risk groups compared to survivors (*p* < 0.05). In the deceased group, 48.3% of deaths (14 patients) were attributable to TRM, while the remaining 51.7% (15 patients) were attributable to disease progression. As for the specific causes that contributed to TRM, two patients (14.3%) died of GVHD, eight (57.1%) of infectious complications, and four (28.6%) of transplant-related organ toxicity. The mean values of TA showed statistically significant differences when comparing the different causes of death (*p* < 0.0001) (Figure 3). Comparing patients by disease risk, there is a statistically significant difference between the mean TA in survivors and patients deceased for disease relapse (*p* < 0.005). In contrast, we did not find significancy when comparing the mean TA in survivors and patients deceased for TRM, in both the high- and low-risk groups (*p* > 0.05).

### 3.4. Relationship between Maximum TA Values and Early Transplant-Related Complications

The patients with a TA above 374 U/L had severe mucosal damage (grade III–IV) more commonly than the other group (83% versus 25%, *p* < 0.0001). No differences were detected in the incidence of pulmonary, renal, and neurological TBI-related toxicity, febrile neutropenia or sepsis, fungal and virus infections, veno-occlusive disease, and I–II-grade hepatic toxicity. We observed a higher incidence of III–IV-grade hepatic toxicity in patients with a TA above 374 U/L (18% and 56%, respectively; *p <* 0.05).

### 3.5. Relationship between Maximum TA Values and Late Transplant-Related Complications

We found no differences between the group with a TA above 374 U/L and the group with a TA below 374 U/L, in the onset of long-term transplant-related complications, such as chronic GVHD, hypothyroidism, pancreatic insufficiency, cataract, and growth hormone deficiency.

We evaluated the maximum TA values’ diagnostic performance in predicting death by leukemia relapse, obtaining the specific AUC–ROC curve (Figure 4A). The maximum TA values showed a high predictive performance in identifying disease progression-related deaths (AUC = 0.865; 95% CI = 0.77–0.93; *p* < 0.0001). The cut-off level of 374 U/L was both highly sensitive (80%) and specific (88.9%). The distribution of TA values is shown in Figure 4B.

### 3.6. Relationship between Maximum TA Values and Inflammation Status

Forty-seven patients (81%) of the group with a TA above 374 U/L, and 15 patients (75%) of the other group, underwent cytokine assays at the end of the TBI treatment. The majority of the various pro-inflammatory mediators that were analyzed had an abnormal concentration in both the groups. We found no statistically significant differences for all the cytokines and chemokines that were evaluated between the two groups, except for tumor necrosis factor (TNF)-α, interleukin (IL)-6, and RANTES (regulated upon activation, normal T cell expressed, and secreted) (*p* > 0.05). In fact, in the second group, the concentrations of TNF-α (mean value 144.5 pg/mL), IL-6 (mean value 133.1 pg/mL), and RANTES (mean value 39573.9 pg/mL) were significantly above the reference range and higher than those observed in the patients of the first group (*p* < 0.0001, *p* < 0.0001, and *p* < 0.001, respectively). The relative differences in the IL-6, TNF-α, and RANTES post-TBI concentrations between the two groups are displayed in Figure 5.

## 4. Discussion

In this study, a rapid increase in TA levels was identified in 91% of the patients who underwent total body irradiation, and 12% of the whole population had very high amylasemia (>500 U/L). The explanation for hyperamylasemia, post TBI, is rather simple. In the human body, different tissues take different times to express radiation damage. While acute-responding tissues have high stem cell activity and a high regenerative capacity, and express their damage quickly, non-proliferative tissues can express their radiation damage with a delay of months [22,23]. The pancreas and the salivary glands belong to the tissue group with high radiosensitivity. Despite being made by secretory cells with a slow turnover, they release a massive number of secretory granules that are rich in proteolytic enzymes, during the radiation-related destruction of serous cells [24]. Therefore, salivary amylase secretion rapidly increases within a few hours after irradiation, and reaches its peak within 12–36 h [25]. The hematopoietic tissue is also an acute-responding tissue, whose delicate balance can be severely damaged by radiations. This is, indeed, the necessary condition, thanks to whom TBI is an effective procedure in HSCT after ALL, before the stem cells’ infusion.

Based on these premises, the occurrence of hyperamylasemia has already been studied in the past. Previous studies described parotid amylase as a possible “biological dosimeter” to identify an external overexposure to radiation [26]. Interestingly, our study does not demonstrate any correlation between either single-dose TBI or fractioned-dose TBI protocols and the degree of parotid response to irradiation damage. Ultimately, in an era in which irradiation protocols and machineries are extremely sophisticated, we can assume that different TBI protocols exert a comparable biological effect on the human body, and we cannot explain the wide range of TA levels that we found in our population. Since the human response to radiation is determined not only by the amount of radiation itself, but also by an interindividual, primarily genetic, predisposition to radiation damage, we investigated whether TA could represent a reliable marker of individual radiosensitivity [27,28,29].

However, by analyzing the early and long-term treatment-related complications, we did not find a relationship between the rise in TA values and clinically expressed radiation damage, both in acute and in late-responding tissues, except for severe mucosal-damaged digestive system and III–IV-grade acute liver injury. The relationship between mucosal and parotid irradiation damage is intuitive, because both are early responding tissues. Acute liver injury is most likely attributable to chemotherapy medications, which are part of the myeloablative pre-transplant conditioning.

The most interesting result that emerged from our study is the strong relationship between the cumulative incidence of death and irradiation-related TA values. Surprisingly, disease progression-related mortality was the most common cause of death in patients with a TA above 374 U/L. These differences were not affected by disease status at the transplant or the type of donor. Since both the salivary and hematopoietic tissues are particularly sensitive to irradiation, we expected patients with a higher TA after TBI to be more prone to tissue toxicity and, therefore, to TRM, rather than leukemia relapse. In fact, although leukemia cells’ response to radiations ranges from remarkable radiosensitivity to considerable intrinsic radioresistance, most pediatric acute lymphoid leukemias are susceptible to TBI. In other words, the severe damage that is suffered by the bone marrow should also negatively affect the leukemic stem cell (LSC) microenvironment. Multiple studies inferred the anti-leukemic activity of TBI. For example, the trial FORUM showed an incidence of relapse that was almost double after chemotherapy-based conditioning, compared to TBI [30].

A possible explanation for the high incidence of death by disease progression in the group with TA above 374 U/L, could be the TBI-related bone marrow (BM) niche damage. Specialized stromal niches are one of the most important bone marrow microenvironment elements. The function of these niches is to support HSC self-renewal and multipotency [31]. TBI severely damages the BM stroma with its hematopoietic niches. Trabecular bone volume loss and microstructure damage are present as early as one week after irradiation [32]. It is estimated that 90% of the irradiated clonogenic bone marrow stroma progenitor cells may be permanently lost, or may lose the multi-lineage differentiation capacity [33]. In consideration of this, a loss of stroma function would prevent successful HSC engraftment, and therefore delay the recovery of innate and adaptive immunity.

The main aim of allogeneic HSCT is to activate the donor’s alloreactive immune cells against the patient’s leukemia, which is the immune process known as the graft-versus-leukemia effect [34]. In the interaction between the donor lymphocyte and LSC, a crucial role is held by stromal cells that induce a functional and efficient lymphocyte homing [35]. As a consequence of the excessive radiation-related damage of the stromal niche, the donor lymphocytes would not migrate and correctly express their cytotoxic effect on residual LSCs. These LSCs can slowly induce the creation of “their” protective niches, facilitated by the secretion of tolerogenic cytokines, such as CCL3, and proliferate until complete relapse, without a correct immune surveillance mediated by donor lymphocytes [36,37]. Ultimately, severe elevation of TA after TBI would suggest an excessive exposition to radiation that causes damage to the hematopoietic niche, with impairment of the correct homing of the donor lymphocyte, leading, in the long run, to relapse of the disease.

Another noticeable consideration from our study is the role that the systemic inflammatory response that affects most tissues, due to whole-body radiation, may play in favoring cancer cells survival [38]. Inflammatory cytokines, such as IL-1, IL-6, IL-17, and TNF-α, are known to be highly elevated within 24 to 48 h of radiation exposure [32,39]. Our data show that the concentrations of TNF-α, IL-6, and RANTES are significantly higher in the group with TA values > 374 U/L. The role of these cytokines in bone resorption and inflammatory disorders is widely described [40,41]. During resorption, the bone delivers numerous growth factors that are stored in the bone matrix. The released cytokines may render the bone microenvironment particularly favorable to cancer cell survival [42,43].

Of course, this study has some limits. First, this is a retrospective consecutive case series of subjects collected from a large time interval, and the number of patients with a very high TA is overall limited. However, the TBI protocols, except for the switch from single high-dose administrations to fractionated regimens, have substantially remained the same in the last 20 years. TBI’s progression has been more about the quality of the technologies, and thus of the machines that provide radiations, rather than the draft of new delivery protocols. Moreover, the largest cohort was needed to achieve a reasonable degree of statistical significance.

Remarkably, this is the first study evaluating irradiation-induced TA values’ performance in identifying highly radiosensitive individuals, with a significant influence on the incidence of overall survival and, in particular, disease relapse.

## 5. Conclusions

Despite the well-known individual heterogeneity in radiation susceptibility, TBI protocols have not yet considered it [21]. Our study suggests that TA might not only be a potential indicator of radiosensitivity, but also a marker of increased risk of leukemia relapse. Furthermore, our study did not correlate with TRM, and could not predict the incidence of long-term damage and radiation consequences. However, further studies on wider populations are needed to confirm this possible relationship in subjects who undergo TBI prior to HSCT. Lastly, new in vitro studies need to better investigate the role of the stromal microenvironment after radiation exposure.

## Figures and Tables

**Figure 1 jcm-10-03834-f001:**
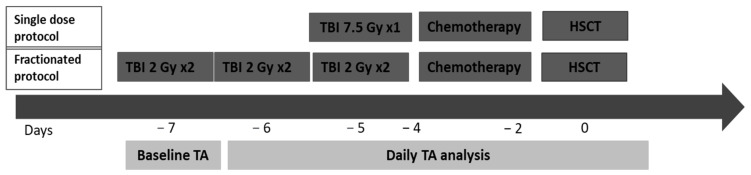
Timeline of total amylase (TA) evaluation. TA was analyzed at baseline, immediately before the TBI procedure, and every day until TA values normalized. Patients have received a standard dose of 12 Gy in three days delivered in six fractions or a single dose of 7.5 Gy in a single day followed by the chemotherapy part of conditioning according to the chosen protocol.

**Figure 2 jcm-10-03834-f002:**
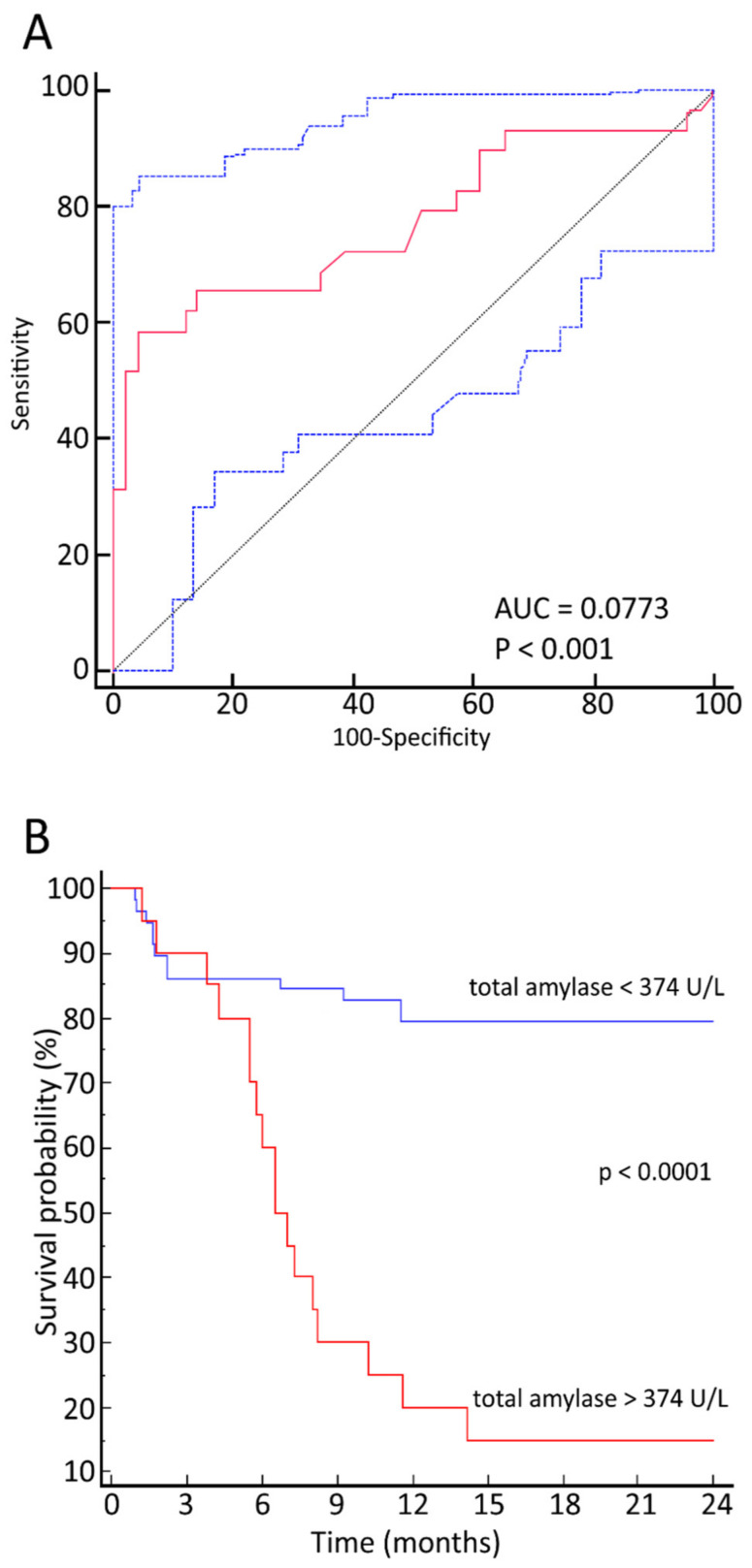
(**A**) Receiver operative characteristics curves of total amylase diagnostic performance in predicting the overall survival after hematopoietic stem cell transplantation with total body irradiation-based conditioning. (**B**) Kaplan–Meier curves for overall survival of patients with total amylase values below 374 U/L (blue line) and above 374 U/L (red line).

**Figure 3 jcm-10-03834-f003:**
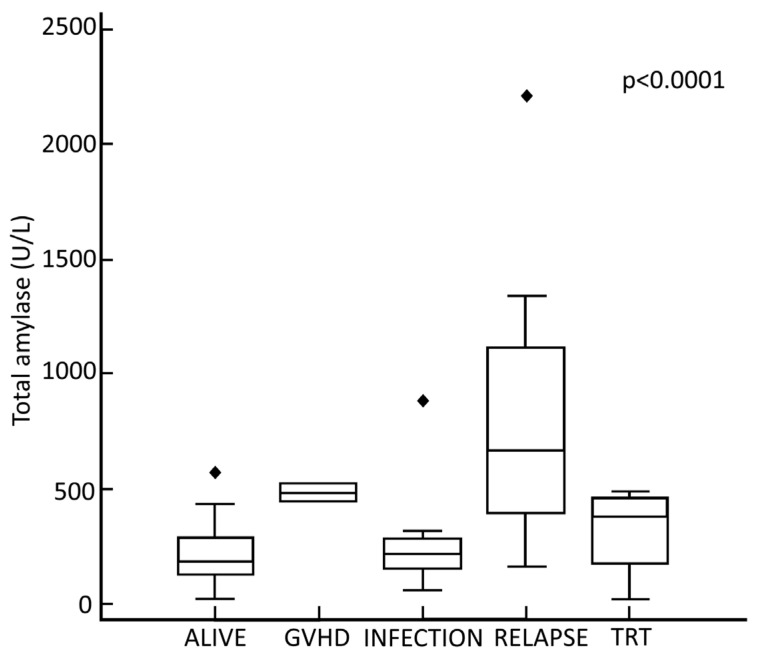
Box-and-whisker plot of maximum total amylase concentration in patients with different transplant-related outcomes. Box plots showing the median (line), upper and lower quartiles (box), and 5% and 95% limits (lines extending from the box). The outcomes are shown on the x-axis. From left to right of the axis, the following categories apply: surviving patients, dead of graft-versus-host disease (GVHD), dead of infection, dead of leukemia relapse, dead of transplant-related toxicity (TRT). The highest TA value was observed in the group of patients who died from a leukemic relapse (673 U/L), followed by death due to GVHD (490 U/L) and by transplant-related organ toxicity (386 U/L). The lowest TA value was detected in the group of patients who died from infectious complications (215 U/L). A Kruskal–Wallis test confirmed the significant difference in TA values of confronted groups (*p* < 0.000061).

**Figure 4 jcm-10-03834-f004:**
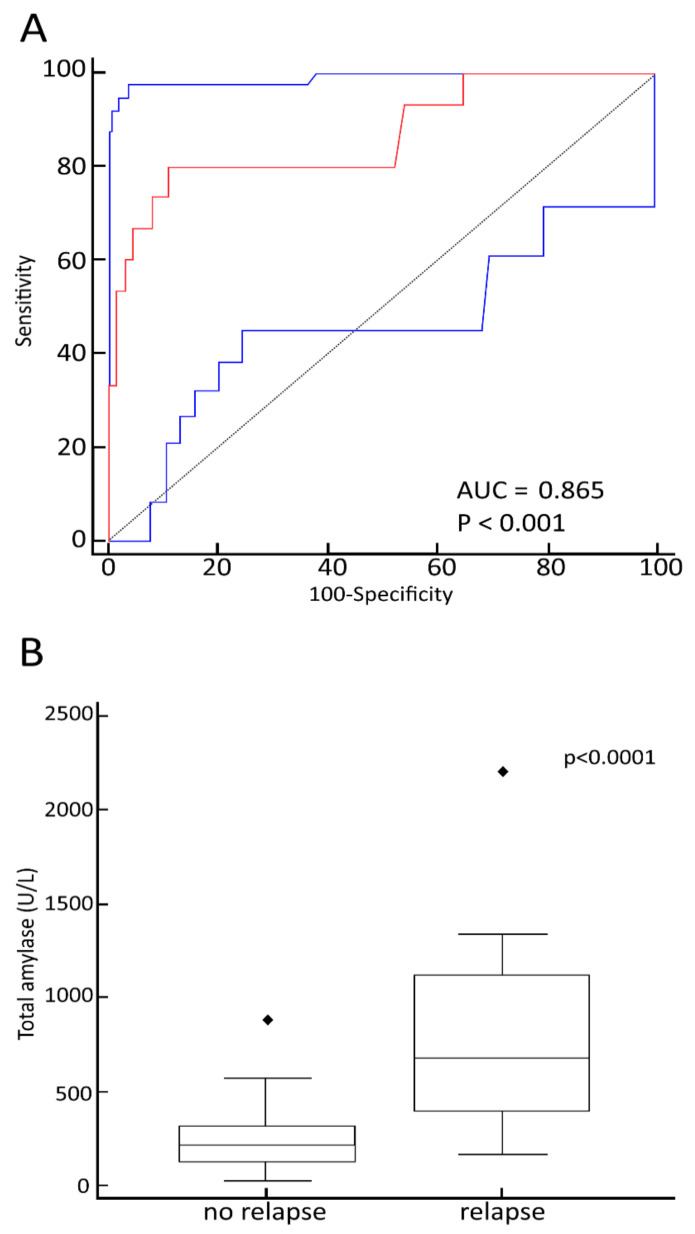
(**A**) Receiver operative characteristics curves of total amylase diagnostic performance in predicting leukemia relapse-related death after hematopoietic stem cell transplantation with total body irradiation-based conditioning. (**B**) Box-and-whisker plot of maximum total amylase values of recurrence-free patients and patients with disease recurrence.

**Figure 5 jcm-10-03834-f005:**
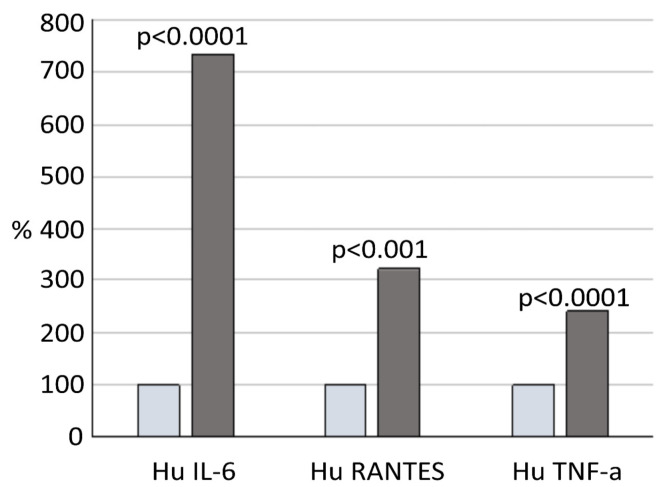
Graphical presentation of relative differences in proinflammatory cytokine values after total body irradiation between the patients with total amylase values below and above 374 U/L. Light-blue column shows values of low TA level group. Grey column shows values of the high TA level group. Abbreviations: Hu IL-6—human interleukin 6, Hu RANTES—human regulated upon activation normal T cell expressed and secreted cytokine, Hu TNF-α—human tumor necrosis factor α, TA—total amylase.

**Table 1 jcm-10-03834-t001:** Characteristics of the ALL patients at transplant.

Pre-Transplant Baseline Charatteristics	Whole Cohort
Number of patients (%)	78 (100)
Sex (%):	
Males	49 (62.8)
Females	29 (37.2)
Age at transplant, years (mean (SD)	10.4 (4.7)
Disease stage at transplant, number (%): *	
Early	21 (26.9)
Late	57 (73.1)
Donor type, number (%):	
HLA-matched related	26 (33.3)
HLA-matched unrelated	33 (42.3)
Haploidentical	19 (24.4)
TBI protocol, number (%):	
12 Gy	57 (73.1)
7.5 Gy	21 (26.9)
TBI-associated chemotherapy, number (%):	
Thiotepa + cyclophosphamide ± ATG	64 (82.1)
Cyclophosphamide ± ATG	8 (10.2)
Fludarabine + thiotepa ± ATG	6 (7.7)
Dose-rate, cGy/min (mean (±SD):	
12 Gy protocol	14.0 (2.0)
7.5 Gy protocol	18.7 (1.7)
Variation in entrance dose, % (mean (±SD):	
12 Gy protocol	1.5 (1.0)
7.5 Gy protocol	−0.9 (1.9)
Baseline serum amylase value, U/L (mean (±SD):	
Total	35.2 (11.4)
Pancreatic	13.6 (7.1)

ALL = acute lymphoblastic leukemia; SD = standard deviation; TBI = total body irradiation; ATG = anti-thymocyte globulin. Serum total amylase normal range 28–100 U/L; serum pancreatic amylase normal range 8–53 U/L. * Disease stage was defined according to previously published classification [21].

## Data Availability

The datasets used and/or analyzed during the current study are available from the corresponding author on reasonable request.

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
