# Peer review of "Post-Irradiation Hyperamylasemia Is a Prognostic Marker for Allogeneic Hematopoietic Stem Cell Transplantation Outcomes in Pediatric Population: A Retrospective Single-Centre Cohort Analysis"

_jcm, 2021, doi:10.3390/jcm10173834_

Round 1

Reviewer 1 Report

Retrospective single study. The authors investigated the relationship between total amylases (TA) and transplant-related outcomes in pediatric recipients with a diagnosis of ALL after TBI. Indeed, this is the first study evaluating irradiation-induced TA values and their impact on outcome in pediatric ALL patients. Yet, there are several concerns and issues:

The biological  risk stratification for ALL as well as the exact type of the disease is completely missing!! This is mandatory to put the results of TA and its impact on outcome into context.

The main aim of the work is the impact of TA on outcome. Thus, why were patients with documented abnormal total amylase (TA) and pancreatic α-amylase levels one month before HSCT excluded? How many were excluded?

In patients under the age of 2, TBI was omitted. N?

Reasons of the two different TBI regimens (standard dose of 12 Gy, delivered in 6 fractions; and the 7.5 Gy in a single dose) need to be given.

Exant time points of Serum TA measurement (median; range) for all measurements: before (what is baseline?), during, and after the TBI treatment.

Time points of measurement of TBI-related inflammatory status (median; range) (see above) and their correlation tot he TA measurements.

The version used oft he National Cancer Institute common toxicity criteria (Line 134) needs to be mentioned.

Was a multivariate analysis performed? This needs to be clearly stated in the statistical part as well as in the discussion section. 

The authors state that OS after HSCT at last follow-up was 62.8% (49 patients). No demographic or transplant-related predictors such as gender, age at transplant, donor type, graft source, TBI protocol, or TBI-associated chemotherapy were related to OS. What about the biological features of the disease such as karyotype? What ist he median (range) follow-up time?

Overall, the results of the measurements need to be mentioned with precise time-points, medians (ranges), and number of patients: examples:

- Which maximum TA values were compared. Those before or after TBI or both and what was the number of patients in each of the two TBI regimens where values were available?

- Which maximum TA values predicted the post-transplant OS. Those before or after TBI or both?

- Under Relationship between maximum TA values, status of disease at transplant and causes of death (Line 196), only percentages without numbers are given. To draw statistocal conclusions from sub-group anylyses from a total of 29 patients is problematic. Exact numbers need to be mentioned!

The discussin is too long. Indeed, strating by Line 304, the discussion has little – if at all- to do with the work done.

In the discussion, the authors write that a possible explanation of a high incidence of disease progression death in group with TA above 374 U/L could be the TBI-related bone marrow (BM) niche damage. They do not explain why this niche damage should be associated with TA?

The authors must try to explain the relationship between TA values and their findings!

The data as currently presented do not allow the conclusion made regarding the correlation of TA (again at which time point?) with leukemia relapse incidence. All biological risk factors and all other variables need to be considered in the statistical analysis with a strong multivariate analysis.  The conclusions must be re-phrased.

Author Response

Point by point review, reviewer 1

Retrospective single study. The authors investigated the relationship between total amylases (TA) and transplant-related outcomes in pediatric recipients with a diagnosis of ALL after TBI. Indeed, this is the first study evaluating irradiation-induced TA values and their impact on outcome in pediatric ALL patients. Yet, there are several concerns and issues:

The biological risk stratification for ALL as well as the exact type of the disease is completely missing!! This is mandatory to put the results of TA and its impact on outcome into context.

A biological risk stratification has been made and it has been reported in table 1 (demographics), according to the paper by Gratwohl A et al “Risk score for outcome after allogenic hematopoietic stem cell transplantation. Cancer 2009;115(20):4715-4726.” (reference n.45)

The main aim of the work is the impact of TA on outcome. Thus, why were patients with documented abnormal total amylase (TA) and pancreatic α-amylase levels one month before HSCT excluded? How many were excluded?

The work aim is to identify a possible relationship between post-irradiation high amylase levels and HSCT outcomes. If a patient had high level of amylase before HSCT, he certainly would have had have high enzymatic titers before TBI. We did not include patients with baseline elevation of TA because their TA elevation was not irradiation-related, and consequentially, not of interest to the study. 

In patients under the age of 2, TBI was omitted. N?

Patients under the age of two were excluded from the study because chemotherapy-only conditioning regimes are commonly preferred for children under 2–4 years old (1). For example, the current FORUM trial uses 4 years as a cut-off for chemotherapy-only versus TBI-based conditioning (2). This is stated in the manuscript: “In patients under the age of 2, TBI was omitted, according to the national protocol for ALL of the Italian Association of Pediatric Hematology and Oncology (AIEOP).”

1) Hoeben BAW, Pazos M, Albert MH, et al. Towards homogenization of total body irradiation practices in pediatric patients across SIOPE affiliated centers. A survey by the SIOPE radiation oncology working group. Radiother Oncol. 2021 Feb;155:113-119.

2) Peters C, Schrappe M, von Stackelberg A, et al. Stem-cell transplantation in children with acute lymphoblastic leukemia: A prospective international multicenter trial comparing sibling donors with matched unrelated donors-The ALL-SCT-BFM-2003 trial. J Clin Oncol. 2015 Apr 10;33(11):1265-7

Reasons of the two different TBI regimens (standard dose of 12 Gy, delivered in 6 fractions; and the 7.5 Gy in a single dose) need to be given.

In recent years, various TBI regimens have been suggested for childhood leukemia. While the 7.5 Gy single-dose irradiation was more common in the past, the 12 Gy fractioned-dose irradiation is widely used nowadays, and represent the standard protocol in various countries, such as the UK (1-2). The 12 Gy fractioned dose is generally preferred because it’s more tolerable by the patients in terms of discomfort and side effects.

We switched from a regimen to another as suggested by the international guidelines.

1) Bushhouse S, Ramsay NK, Pescovitz OH, et al. Growth in children following irradiation for bone marrow transplantation. Am J Pediatr Hematol Oncol. 1989 Summer;11(2):134-40. 

2) Radiotherapy dose fractionation, third edition. The Royal College of Radiologists. 2019;9:65-75.

Exant time points of Serum TA measurement (median; range) for all measurements: before (what is baseline?), during, and after the TBI treatment.

“Baseline” means analysis of TA immediately before TBI. Baseline TA and pancreatic amylase values are reported in table 1. “During” means either after the first day of TBI (in patients who underwent the single dose protocol) or after the third TBI session (in patients who underwent the fractioned dose protocol). In order to make this concept clearer for the reader, we better described this part in the manuscript. “After” indicates that, in all the patients in which a high titer of TA was found after the TBI protocol, TA was controlled until HSCT or until complete normalization that, eventually, occurred to all the subjects. We clarified this aspect in the dedicated part of the article as suggested.

Time points of measurement of TBI-related inflammatory status (median; range) (see above) and their correlation tot he TA measurements.

As reported in the materials and methods section, we considered only the cytokine values determined at the end of TBI treatment. Therefore, the dosage was performed after the single day of TBI (in patients who underwent the single dose protocol) or after the sixth TBI session (in patients who underwent the fractioned dose protocol). Instead of reporting all the various mean values and ranges off all the cytokines analyzed (which are more than 20), we drew figure 4, in which only the statistically significant molecules (and their different dosage between high and low TA subjects) is reported. This choice was made for clarity purposes. However, as suggested by the reviewer, we added the dosage of the relevant cytokines in discussion section.

The version used of the National Cancer Institute common toxicity criteria (Line 134) needs to be mentioned.

Changes have been made accordingly.

Was a multivariate analysis performed? This needs to be clearly stated in the statistical part as well as in the discussion section. 

Due to the small dimension of the study population, we did not perform a multivariate analysis.

The authors state that OS after HSCT at last follow-up was 62.8% (49 patients). No demographic or transplant-related predictors such as gender, age at transplant, donor type, graft source, TBI protocol, or TBI-associated chemotherapy were related to OS. What about the biological features of the disease such as karyotype? What ist he median (range) follow-up time?

Karyotype anomalies are part of the pre-transplant biological risk stratification. If the reviewer is referring specifically to Philadelphia chromosome-positive (Ph+) acute lymphoblastic leukemia (ALL), none of subjects was Ph+.  Follow up time is reported in the discussion section: “The median follow-up was 4.3 years (range 1.08 – 12.83 years)”

Overall, the results of the measurements need to be mentioned with precise time-points, medians (ranges), and number of patients: examples:

- Which maximum TA values were compared. Those before or after TBI or both and what was the number of patients in each of the two TBI regimens where values were available?

We compared the TA values after TBI. TA prior to TBI was considered only as exclusion criteria, to rule those who already had abnormal values. In these subjects we would have not been able to make any sort of correlation with post irradiation values.

- Which maximum TA values predicted the post-transplant OS. Those before or after TBI or both?

All the results described in the paper are referred to post-irradiation TA.

- Under Relationship between maximum TA values, status of disease at transplant and causes of death (Line 196), only percentages without numbers are given. To draw statistocal conclusions from sub-group anylyses from a total of 29 patients is problematic. Exact numbers need to be mentioned!

As requested, all the exact numbers have benne added to the paragraph.

The discussion is too long. Indeed, strating by Line 304, the discussion has little – if at all- to do with the work done.

As suggested by the reviewer, the first part of the discussion has been rewritten

The key elements of the discussion are the following:

  • Elevated TA are a common finding in patients who underwent TBI prior to HSCT (again, we considered post irradiation hyperamylasemia)
  • TA are elevated because the pancreatic and salivary tissue, like the hematopoietic tissue, are acutely responsive to radiations
  • The interindividual variability is not determined by different TBI protocols, but by intrinsic genetic factors
  • Hyperamylasemia was not connected with clinically expressed radiation damage and TRM, but rather with death by leukemia relapse.
  • A possible explanation is the connected to the role of the TBI-related bone marrow (BM) niche damage. Patients with higher TA may have a higher susceptibility to radiations. In these patients, TBI may permanently insult the stromal niche, which plays a key role in guaranteeing a successful homing of donor lymphocyte and thus in activating the graft-versus-leukemia effect. This implies that in this subgroup of patients excessive TBI exposition could cause more harm than benefit.
  • We found an increased concentrations of TNF-α, IL-6, and RANTES in patients with high TA values. Some of them may play a negative effect on the stromal microenvironment that can potentially favors cancer cell survival.

explanation of a high incidence of disease progression death in group with TA above 374 U/L could be the TBI-related bone marrow (BM) niche damage. They do not explain why this niche damage should be associated with TA?

We added a small paragraph to describe this connection. I will report it here below:“Ultimately, severe elevation of TA after TBI would suggest an excessive exposition to radiation that cause a damage on hematopoietic niche with impairment of the correct homing of donor lymphocyte, leading in the long-run to relapse of the disease”

The authors must try to explain the relationship between TA values and their findings!

This part has also been changed in the discussion section.

The data as currently presented do not allow the conclusion made regarding the correlation of TA (again at which time point?) with leukemia relapse incidence. All biological risk factors and all other variables need to be considered in the statistical analysis with a strong multivariate analysis.  The conclusions must be re-phrased.

Conclusions were changed as requested. We highlighted that further study on TA are needed to confirm our results.

Reviewer 2 Report

In this manuscript, the authors analyze the clinical impact of levels of total amylases (TA) in acute lymphoblastic leukemia (ALL) cases who had allogenic hematopoietic stem cell transplantation (HSCT) under total body irradiation (TBI) contained regimens. Although it is understandable that identification of laboratory markers which predict clinical course is important in this field, there are some flaws in the manuscript.

1) First, it is difficult to understand entire study design (e.g. Timings of TA analysis and its relationships to TBI and other therapeutic drugs). A summary figure depicting the entire design will be welcomed to understand and follow this study.

2) In Figure 1B, it is unclear about the definition of overall survival (OS). Since the authors stratified the cases into two groups based on the maximum TA levels in each case, the OS should be evaluated from the date of testing which showed the maximum TA level.

3) It is unclear which TA data were analyzed in Figures 2 and 3. Although Figure 2 showed that the levels were higher in relapsed cases, does this mean that relapsed cases had already had elevation of TA or TA was elevated proceeded prior to the relapse.

4) In Figure 4, what is the control for the “relative” differences? Also, which samples were used for the cytokine analyses?

Author Response

Point by point review, reviewer 2

In this manuscript, the authors analyze the clinical impact of levels of total amylases (TA) in acute lymphoblastic leukemia (ALL) cases who had allogeneic hematopoietic stem cell transplantation (HSCT) under total body irradiation (TBI) contained regimens. Although it is understandable that identification of laboratory markers which predict clinical course is important in this field, there are some flaws in the manuscript.

1) First, it is difficult to understand entire study design (e.g. Timings of TA analysis and its relationships to TBI and other therapeutic drugs). A summary figure depicting the entire design will be welcomed to understand and follow this study.

Study design tried to correlate how  and salivary glands damage may be associated to precocious bone marrow stoma damage and consequentially an inefficient homing of donor lymphocytes in bone marrow that guarantee a sustained eradication of leukemic stem cell. Elevation of TA was measured before TBI, as a baseline, and every day during TBI until the day of HSCT infusion. Between the measured values, we chose the highest as the analyzed variable. We explain this aspect better in the article's dedicated section and add a summary figure as suggested by the reviewer.

2) In Figure 1B, it is unclear about the definition of overall survival (OS). Since the authors stratified the cases into two groups based on the maximum TA levels in each case, the OS should be evaluated from the date of testing which showed the maximum TA level.

Overall survival was evaluated after HSCT during the follow-up of the study, evaluating all the causes of death. Considering that all the highest of TA were before HSCT, overall survival after HSCT is comparable to OS after the date of the testing with the maximum TA level. We clarify better this aspect in the section. 

3) It is unclear which TA data were analyzed in Figures 2 and 3. Although Figure 2 showed that the levels were higher in relapsed cases, does this mean that relapsed cases had already had elevation of TA or TA was elevated proceeded prior to the relapse.

We measured TA elevation during or immediately after TBI conditioning with a progressive normalization in the following days. Consequently, we analyzed the incidence of relapse during follow-up in patients who had previously elevated TA. The study's core finding is that patients who had a higher elevation of TA during conditioning have a higher incidence of relapse after HSCT.

4) In Figure 4, what is the control for the “relative” differences? Also, which samples were used for the cytokine analyses?

In figure 4 we assessed the difference of IL-6, TNF and RARES in the group with mild elevation of amylases and the group with severe elevation of amylases. As control, we use the mild elevation group (under 374 U/L) and considered the relative proportion with the severe elevation group. For example, IL-6 elevation was about seven times higher in the severe TA elevation group compared to the mild TA elevation group. Samples used were obtained from the same blood workup used to measure TA daily. In addition, a dedicated laboratory analysed cytokine levels on serum samples. We added in the article also mean values of measured cytokines to give a better exhaustiveness on the matter. 

Round 2

Reviewer 1 Report

The authors responded adequately to my concerns

Author Response

No changes were requested.

Reviewer 2 Report

The authors have responded the concerns raised by the reviewer. I think the responses will make the manuscript clearer, but the authors did not reflect a part of the responses to the manuscript (e.g. It is still unclear about the definition of OS and the relative expressions of the cytokine levels in the MANUSCRIPT though the authors responded them in the response letter). Similarly, there is no definition of the colors for the plots in Figure 5 though now I can understand the meanings from the author's reply.  This point will lead to the reproducibility of the study. 

Author Response

Answer to reviewer 2

The authors have responded the concerns raised by the reviewer. I think the responses will make the manuscript clearer, but the authors did not reflect a part of the responses to the manuscript (e.g. It is still unclear about the definition of OS and the relative expressions of the cytokine levels in the MANUSCRIPT though the authors responded them in the response letter). Similarly, there is no definition of the colors for the plots in Figure 5 though now I can understand the meanings from the author's reply.  This point will lead to the reproducibility of the study. 

We better explain the concept of overall survival: we analyze the survival of recruited patients until the latest follow-up (mean 4.2 years). We also add the mean values of cytokine levels on the manuscript (lines 268-271). Finally, we better describe the meanings of the plots in figure 5 as suggested by the reviewer.